# The effects of chronic diseases on plutonium urinary excretion in former workers of the Mayak Production Association

Klara G. Suslova[1]*, Alexander V. Efimov[1], Alexandra B. Sokolova[1], Bruce A. Napier[2]*, Scott C. Miller[3]*

**1** Southern Urals Biophysics Institute (SUBI), Ozyorsk, Chelyabinsk Region, Russian Federation, **2** Pacific Northwest National Laboratory, Richland, Washington, United States of America, **3** Radiobiology Division, Department of Radiology and Imaging Science, School of Medicine, University of Utah, Salt Lake City, Utah, United States of America

\* Suslova@subi.su (KGS); Bruce.Napier@pnnl.gov (BAN); Scott.Miller@hsc.utah.edu (SCM)

**Data Availability Statement:** Access to the data is available only upon request in compliance with the "Data Access Agreement" from the Joint Committee on Radiation Effects Research

## Abstract

The radiochemical analysis of plutonium activity in urine is the main method for indirect estimation of doses of internal exposure from plutonium incorporation in professional workers. It was previously shown that late-in-life acute diseases, particularly those that affect the liver, can promote accelerated rates of release of plutonium from the liver with enhanced excretion rates. This initial study examines the relationships of some chronic diseases on plutonium excretion as well as the terminal relative distribution of plutonium between the liver and skeleton. Fourteen cases from former workers at the Mayak Production Association (Mayak PA) who provided from 4–9 urine plutonium bioassays for plutonium, had an autopsy conducted after death, and had sufficient clinical records to document their health status were used in this study. Enhanced plutonium excretion was associated with more serious chronic diseases, including cardiovascular diseases and other diseases that involved the liver. These chronic diseases were also associated with relatively less plutonium found in the liver relative to the skeleton determined by analyses conducted after autopsy. These data further document health conditions that affect plutonium biokinetics and organ deposition and retention patterns and suggest that health status should be considered when conducting plutonium bioassays as these may alter subsequent dosimetry and risk models.

## Introduction

The Mayak Production Association (Mayak PA) constructed in 1948 was the first facility to produce plutonium primarily for nuclear weapons in the former Soviet Union. During the first decade of operation (1948–1958), a considerable number of workers were exposed to high levels of plutonium, primarily through inhalation of industrial aerosols. Exposures continued in later years but were greatly reduced with the introduction of more effective hygiene measures [1]. The Mayak Worker cohort contains more than 25,000 workers and studies have shown dose-related increases in cancers particularly in organs where plutonium is

(JCCRER). The JCCRER is a committee established by a bilateral agreement between the U.S. and Russian Governments, signed by the U.S. Secretary of State and the Foreign Minister for the Russian Federation. Under this agreement, requests for data can be directed to Dr. Sergey Romanov, Director of the Southern Ural Biophysics Institute, Ozyorsk, Russia at Romanov@subi.ru.

**Funding:** The Joint Coordinating Committee for Radiation Effects Research, jointly funded by the United States Department of Energy and the Federal Medical Biological Agency of the Russian Federation, provided salary support to the authors under Project Number AU 00000902. The funders had no role in study design, data collection and analysis, decision to publish, or preparation of the manuscript.

**Competing interests:** The authors have declared that no competing interests exist.

preferentially deposited, specifically lung, liver, and bone [2–6]. The Mayak Worker cohort has become a major source of information on the risks of longer-term occupational exposures to radiation [7].

Epidemiological studies that predict the risks associated with radiation doses from internally deposited plutonium are dependent on dose estimates to specific organs and tissues being obtained. In many cases, dosimetry estimates are made during life from bioassay measurements of plutonium excreted in urine. From the excretion rates, systemic and organ dosimetry estimates can be obtained from biokinetic and dosimetry models. Such models that have been applied to exposed Russian nuclear workers include the Mayak Worker Dosimetry System-2008 (MWDS-2008) [8], MWDS-2013 [9] and MWDS-2016 [10] models. Prior analyses of data obtained from workers at the Mayak Production Association (Mayak PA), Ozersk, Russia, have demonstrated considerable variability in plutonium excretion rates [11]. While there are a number of possible sources of variability in urine bioassay measurements, as discussed by Kathren and McInroy [12], improved calculation methods for bioassay results have been applied to the recent Mayak worker dosimetry models [13].

Late-in-life diseases, particularly those that affect the liver, are known to affect plutonium excretion rates as well as the relative retention of plutonium between the liver and skeleton. In a group of Mayak workers with advanced, late-in-life liver disease, for example, plutonium excretion rates were greatly elevated over those without liver disease [14, 15]. These studies also showed a correlation between disease severity and a relative decrease in the fraction of systemic (non-pulmonary) plutonium found in the liver with a concomitant increase in the skeleton. It was suggested that as plutonium was being released from the liver, more plutonium entered the circulation and became available for excretion or uptake into skeletal tissues [16]. The diseases that were associated with the greater excretion rates and relative redistribution of plutonium between the liver and skeleton included alcohol-related liver diseases, other diseases associated with fatty degeneration of the liver, and primary and metastatic cancers of the liver [16].

The prior studies cited above considered mostly late-in-life diseases, most of which were acute in nature. The purpose of the present study was to determine if some chronic diseases, particularly those that might affect the liver, have effects on plutonium excretion rates and the relative distribution of plutonium between the liver and skeleton. For this study, a group of Mayak PA workers were selected who had chronic diseases confirmed from clinical records, had multiple plutonium urine bioassays over a range of years, and had an autopsy conducted where terminal plutonium organ contents were measured. These initial results suggest that chronic diseases that involve the liver are associated with longer-term elevated rates of plutonium excretion. Additionally, chronic diseases are associated with less retention of plutonium in the liver relative to that in the skeleton as determined after autopsy.

## Materials and methods

### Cases

The study was done using anonymized records and was approved by the Human Ethics Review Panel of the Southern Urals Biophysics Institute and in compliance with the laws of the Russian Federation. Fourteen de-identified cases were selected from the Mayak PA database of former workers that met the following criteria: 1) They had suitable work histories and clinical records; 2) they had submitted to 4–9 bioassays for urinary excretion of plutonium; 3) their general health and status of chronic diseases during the periods of their bioassays could be estimated from the clinical records; and, 4) a terminal autopsy was conducted with measurements made to estimate total body and organ plutonium contents. The autopsies provided

additional information on the presence and severity of any chronic diseases. The exposure route for the workers was through inhalation of plutonium containing aerosols in the workplace.

The characteristics of the cases are described in Table 1 and include the primary work location(s), years of employment at the Mayak PA, year of death, the total body plutonium burden, and information on the urine bioassays for plutonium that include the number of bioassays, the age at first bioassay and the year range when the bioassays were conducted. The selected individual workers started their work at the Mayak PA from 1949–1964 and finished working from 1956–1992 with employment ranging from 4–28 years. The workers had worked in the plutonium processing plant and/or the radiochemical plant. These plants were described in detail by Vasilenko et al. [1]. Urine bioassay measurements were initially conducted from 1971–1987 and the last bioassay for each worker ranged from 1984–2008. Bioassay results taken within one year of death were not used in this study.

## Determination of health status

Several sources of information were used to reconstruct the health status of the workers during the period that bioassays were conducted. The workers were observed as inpatients in the clinics when the urine bioassays were conducted. Medical examinations were also performed at this time, thus the clinical records corresponded with the individual bioassay measurements. The health information in the records included any clinical diagnoses, status and/or progression of existing disease and information on smoking and alcohol use. Autopsy records were also available and included cause(s) of death, status of prior disease(s), and histopathology findings. The plutonium contents of the entire body and selected organs and tissues were also available. Because it was previously reported that liver diseases can alter the systemic (extrapulmonary) distribution and excretion of plutonium in the Mayak workers [15, 16], particular attention was given to the occurrence and progression of liver diseases or other diseases, such as congestive heart failure [17], known to involve the liver.

**Table 1. Characteristics of selected cases.**

| ID | Work location[a] | Work period | Year of death | Plutonium body content, kBq | Bioassay examinations | | |
|---|---|---|---|---|---|---|---|
| | | | | | Number of bioassays | Age at first bioassay | Years of bioassay examinations |
| 2095 | Pu | 1949–1957 | 2009 | 2.9 | 7 | 46 | 1975–2008 |
| 3108 | Rc | 1953–1957 | 2014 | 1.2 | 7 | 46 | 1972–2006 |
| 3513 | Pu | 1957–1978 | 2009 | 6.2 | 5 | 52 | 1987–2002 |
| 6962 | Pu | 1957–1984 | 2010 | 4.7 | 8 | 47 | 1977–2002 |
| 7790 | Pu | 1957–1984 | 2009 | 4.6 | 4 | 43 | 1983–2008 |
| 8046 | Rc | 1954–1971 | 1989 | 7.0 | 6 | 61 | 1980–1989 |
| 8260 | Pu | 1949–1962 | 1993 | 6.0 | 6 | 63 | 1977–1990 |
| 8686 | Pu | 1949–1960 | 1989 | 7.5 | 6 | 54 | 1972–1988 |
| 6984 | Pu | 1949–1974 | 1986 | 2.7 | 9 | 47 | 1971–1986 |
| 9127 | Pu | 1959–1973 | 1992 | 0.6 | 5 | 62 | 1976–1989 |
| 9277 | Pu+Rc | 1948–1956 | 1988 | 32.0 | 7 | 47 | 1974–1985 |
| 9351 | Rc | 1964–1992 | 1992 | 5.8 | 7 | 36 | 1974–1989 |
| 9418 | Pu | 1953–1960 | 1993 | 14.2 | 6 | 56 | 1978–1993 |
| 9451 | Pu | 1956–1983 | 1986 | 19.4 | 8 | 43 | 1975–1984 |

[a]Work locations: Pu, plutonium plant; RC, radiochemical plant.

The cases were assigned to a "health status" group based on their records and these catego-ries were similar to those used in prior studies [15, 16]. Those cases assigned to the "Healthy" category usually died from acute events or from a late cancer that likely did not affect the liver during the periods that bioassays were conducted. Individuals in this group had a number of other chronic diseases that included: cerebrovascular diseases, acute cerebrovascular events, diabetes, some cardiovascular diseases, gastrointestinal tract diseases, bone tissue dystrophies (osteochondroses) and others. Individuals with significant primary or secondary liver diseases were not included in the "Healthy" status group. Those cases assigned to the "Ill" health status category included cases with chronic or malignant diseases and conditions that appeared to be aggravated by alcohol consumption. Overt liver and acute cardiovascular diseases were not included in this category. The "Severely Ill" health status category included chronic diseases such as cancer metastases to the liver, heart diseases (such as congestive heart failure) and alco-hol-related liver diseases. The cases assigned to these three health status groups are presented in Table 2. One case (#9451) was initially assigned to the "Ill" category but was later reassigned to the "Severely Ill" category due to progressive hepatic cirrhosis. A summary of the character-istics of the health status groups is presented in Table 3.

## Urine bioassays measurements for plutonium

Urine plutonium bioassays were conducted during intervals for each case as indicated in Table 2 and overall the bioassays were conducted from 1971 through 2008. For these assays, workers were typically housed as inpatients for 3 or more days with multiple 24-hour urine samples collected [13]. From 1971 to 1998, plutonium was measured using alpha-radiometic methods. From 1998 to 2008, the assays were done using more sensitive alpha-spectrometry

**Table 2. Diseases recorded in the clinical records during the period of bioassay examinations and assignments to "health status" groups.**

| ID | Bioassay period (yr)[a] | Diseases[b] noted in the clinical records during bioassay examination period | Health Status Group |
|---|---|---|---|
| 2095 | 1975–2008 | Gastrointestinal disease, congestive heart failure, hepatic congestion. | Severely Ill |
| 3108 | 1972–2006 | None | Healthy |
| 3513 | 1987–2002 | None | Healthy |
| 6962 | 1977–2002 | None | Healthy |
| 7790 | 1983–2008 | Gastrointestinal disease, chronic lung disease | Ill |
| 8046 | 1980–1989 | Cardiovascular disease, congestive heart failure, alcohol-related liver disease | Severely Ill |
| 8260 | 1977–1990 | None | Healthy |
| 8686 | 1972–1988 | None | Healthy |
| 8984 | 1971–1986 | Hepatitis, cerebrovascular disease, cancer with late liver metastases | Severely Ill |
| 9127 | 1976–1989 | Stomach cancer, chronic lung disease with later lung cancer, congestive heart failure | Severely Ill |
| 9277 | 1974–1985 | Cerebrovascular disease, gastrointestinal disease | Ill |
| 9351 | 1974–1989 | Alcohol-related liver disease, congestive heart failure | Severely Ill |
| 9418 | 1978–1993 | Cardiovascular disease, pneumonia | Ill |
| 9451 | 1975–1984 | Liver cirrhosis, later stomach cancer | Ill, Severely Ill[c] |

[a]From Table 1.

[b]Particular emphasis on diseases that may have affected the liver.

[c]In Ill group from 1975–1981, then in Severely Ill group from 1983–1984.

**Table 3. Summary of characteristics of the health status groups.**

| Health Status Group | Total number of bioassays | Age range at initial bioassay | Initial bioassay period (y range) | Last bioassay period (y range) |
|---|---|---|---|---|
| "Healthy" | 32 | 44–63 | 1972–1987 | 1987–2006 |
| "Ill" | 23 | 43–56 | 1974–1983 | 1976–1999 |
| "Severely Ill" | 36 | 36–62 | 1971–1983 | 1984–2008 |

methods. The radiochemical procedures and associated limitations and uncertainties for these methods have been described in greater detail [13, 18–20].

## Autopsy measurements for plutonium

Samples were collected from each autopsy case that included lungs, pulmonary lymph nodes and extrapulmonary organs that included liver, bone, spleen, and kidneys for the purposes of radiochemical analyses. Wet ashing techniques were applied to the soft tissues and dry ashing to bones. Prior to 2000, the tissues were prepared and plutonium was measured by alpha-radiometry and after 2000 plutonium was measured by alpha-spectrometry [20, 21]. Usually only selected samples were taken from the skeleton and some soft tissues, including the liver, and total organ weights were not recorded in some of the autopsy records. Thus, the total organ masses were estimated as previously described [8, 21] with total organ masses relative to body weight obtained from "Reference Man" from the International Commission for Radiation Protection (ICRP) [22].

## Determination of excretion coefficients

To correlate plutonium urinary excretion with health status, coefficients for urinary plutonium excretion were determined for each case as of the date of bioassay examination. The excretion coefficients were calculated as previously described [16] relative to the total body content of plutonium, $K_{e\ body}$, and the systemic, non-pulmonary body content of plutonium, $K_{e\ syst}$, measured at autopsy. Briefly, $K_{e\ body}$–coefficient of the nuclide excretion from the body as the ratio of mean plutonium activity in the daily volume of urine $U_m$ to the postmortem total body burden, estimated from the autopsy data.

$$K_{e\ body} = U_m/Q_{body} \tag{1}$$

$K_{e\ syst}$—coefficient of the nuclide excretion from the systemic burden (excluding the lungs and lymph nodes) expressed as a fraction of plutonium systemic content excreted per day:

$$K_{e\ syst} = U_m/Q_{syst} \tag{2}$$

Where: $Q_{body}$ and $Q_{syst}$ are plutonium body content and plutonium systemic content, respectively (in Bq). $U_m$ is the mean content (in Bq) of plutonium in the daily samples collected during each examination period. To calculate the plutonium body burden or plutonium systemic content, $Q_{body}$ and $Q_{syst}$, the plutonium concentration in Bq/g was extrapolated to the organ masses from "Reference Man" [22].

## Partitioning of plutonium between liver and skeleton determined after autopsy

The distribution of plutonium between skeleton and liver was determined as their relative percentage fractions of the systemic (non-pulmonary) plutonium content in organs.

## Statistical methods

The mean excretion coefficient was calculated from the multiple bioassays conducted over time for each worker in each of the health status groups. The data for each of the groups are expressed as the mean ± SD and as geometric means and geometric standard deviations (GSD). Differences between mean values for each of the health status groups were tested for significance using the single-factor analysis of variance (F-test). When statistically significant, pairwise comparisons were done using the least significant difference criterion. Results were considered to be statistically significant when $p < 0.05$.

## Results

### Excretion coefficients

Some of the workers had bioassays conducted near the end of their employment period, but most bioassays were conducted after employment and thus after the period where plutonium inhalations would have likely occurred. As expected, the excretion coefficients were very low but did correlate with overall health status during the period when bioassays were conducted (Table 4). The lowest excretion coefficients, $K_{e\ body}$ and $K_{e\ syst}$, were found in the cases that were assigned to the "Healthy" group.

In those with more severe chronic diseases the excretion coefficients, $K_{e\ body}$ and $K_{e\ syst}$, were progressively greater in the "Ill" group and in the "Severely Ill" group (Table 4). The "Severely Ill" group was assigned those cases with more severe primary and secondary liver diseases that included liver metastases, alcohol related liver diseases, and cardiovascular diseases that affected the liver, such as congestive heart failure that resulted in hepatic congestion (Table 2). Comparing the $K_{e\ body}$ and $K_{e\ syst}$ in the "Healthy" vs. the "Severely Ill" groups, for example, the $K_{e\ body}$ and $K_{e\ syst}$ were almost 3 times greater in the "Severely Ill" group compared with the "Healthy" group (Table 4). The highest single measurement of $K_{e\ body}$ was 3.68 $\times 10^{-5}$ d$^{-1}$ in an individual undergoing chemotherapy (ID 9127 in Table 2). The mean values of $K_{e\ body}$ and $K_{e\ syst}$ were statistically significant between the groups using a single-factor analysis of variance (Table 4).

### Partitioning of plutonium between the liver and skeleton

The relative increases in the $K_{e\ syst}$ associated with increasing severity of chronic diseases (Table 4) were also associated with decreases in the relative distributions of plutonium between the liver and skeleton as determined at autopsy (Table 5). While the variations (standard deviations) of these measurements were rather large, the mean liver/skeleton ratios of the percent of systemic plutonium content decreased with increasing chronic disease severity.

**Table 4. Urinary plutonium excretion coefficients calculated for the entire body ($K_{e\ body}$) and for the systemic, non-pulmonary compartment ($K_{e\ syst}$) for the three health status groups of former Mayak PA workers.**

| Health Status Group | Excretion coefficient | Arithmetic Mean ± SD ($K_e \times 10^{-5}\ d^{-1}$) | Geometric Mean | Geometric SD | Range |
|---|---|---|---|---|---|
| | | | | | Min—Max |
| Healthy | $K_{e\ body}$ | 0.67 ± 0.34[a] | 0.59 | 1.73 | 0.11–1.88 |
| | $K_{e\ syst}$ | 1.11 ± 0.59 [a] | 0.97 | 1.74 | 0.26–2.78 |
| Ill | $K_{e\ body}$ | 1.05 ± 0.51 [a] | 0.95 | 1.53 | 0.41–2.80 |
| | $K_{e\ syst}$ | 1.36 ± 0.75 [a] | 1.15 | 2.00 | 0.11–3.72 |
| Severely Ill | $K_{e\ body}$ | 1.86 ± 0.58 [a] | 1.78 | 1.35 | 0.97–3.68 |
| | $K_{e\ syst}$ | 2.96 ± 1.12 [a] | 2.76 | 1.46 | 1.38–5.74 |

[a]Statistically significant differences in mean values $K_{e\ body}$ and $K_{e\ syst}$ for all three groups were noted at level $p<0.05$ (F-test 52.7).

**Table 5. Correlation of systemic plutonium urinary excretion measured from in-life bioassays and relative plutonium distribution between liver and skeleton determined at autopsy for the three health status groups of former Mayak PA workers.**

| Health Status Group | Number of cases | Years of death | Years of examination | $K_{e\ syst} \times 10^{-5}\ d^{-1}$ (mean ± SD) | Relative distribution, % of systemic content, ± SD | | Liver/Skeleton Ratio, ± SD |
|---|---|---|---|---|---|---|---|
| | | | | | Liver | Skeleton | |
| Healthy | 5 | 1989–2014 | 1972–2006 | 1.11 ± 0.59[a] (32)[b] | 23.9 ±16.7 | 70.4 ± 16.0 | 0.74 ± 0.49 |
| Ill | 4 | 1986–2009 | 1970–1999 | 1.36 ± 0.75 [a] (23) [b] | 18.6 ± 16.3 | 75.5 ± 9.8 | 0.24 ± 0.20 |
| Severely Ill | 6 | 1986–2009 | 1971–2008 | 2.96 ± 1.12 [a] (36) [b] | 13.4 ± 9.5 | 80.4 ± 9.3 | 0.21 ± 0.16 |

[a]Data from Table 4.

[b]Total number of bioassays from Table 3.

## Discussion

The data from this initial study have established that some chronic conditions that occurred during the life of the workers at the Mayak PA were associated with prolonged elevated excretion rates of plutonium. The plutonium excretion rates, relative to the total body contents of plutonium, were progressively greater with disease severity. Additionally, the relative contents of plutonium in the liver relative to the skeleton, measured at autopsy, were less with progressive severity of the chronic diseases.

Acute late-in-life diseases, particularly those that affect the liver, are known to be associated with increased excretion rates of plutonium [14–16]. This was attributed to more plutonium entering the circulation from the liver and being available for excretion in the urine with the remainder being available for deposition in other tissues, particularly the skeleton [16]. This is supported by the autopsy data in the present study where the relative liver/skeleton ratios of plutonium were less with progressive severity of the chronic and acute late-in-life disease, supporting observations from prior studies on the Mayak PA workers [14–16].

In the present study, chronic conditions were identified in the clinical records that were known to be associated with liver diseases. These conditions included alcohol-related liver diseases, cirrhosis, cancers metastatic to the liver, cancer treatments that may have affected the liver, and congestive heart failure. Congestive heart failure, for example, was reported in the clinical records in four (ID 2095; 8046; 9127; 9351, Table 2) of the workers in the "Severely Ill" category and is known to be associated with both the cause and the consequence of certain liver dysfunctions [17, 23].

The relative reductions in liver plutonium content were associated with more serious chronic diseases. Prior studies have associated these changes and differences with histopathologically-defined liver diseases in the Mayak PA workers that included fatty degeneration and hepatocyte dystrophy [16]. Relative reductions in liver contents of fallout plutonium in humans with liver diseases have also been reported [24, 25]. In experimental studies, reductions in liver tissue metals and trace elements have been reported in a model of non-alcoholic fatty liver disease and these reductions were often concomitant with decreases in total liver protein [26]. Various liver diseases and at various stages of disease may also alter biliary excretion of the plutonium and thus fecal excretion rates relative to urinary excretion.

In addition to overt liver disease, alcohol consumption is known to be associated with reductions in liver content of plutonium in humans [16, 25] as well as americium in beagle dogs [27] and baboons [28]. Additionally, ethanol is reported to increase americium excretion

rates from the liver in baboons [28, 29] and thorium in an individual previously injected with the radio-imaging agent Thorotrast with a bioassay conducted the day after moderate alcohol consumption [29]. These observations suggest that in addition to chronic and acute liver disease, the consumption of alcohol in the absence of overt liver disease may also result in elevated excretion rates of radionuclides and should be considered when conducting a bioassay.

The workers in this study were exposed to plutonium compounds during their work at the plutonium and radiochemical plants at the Mayak PA. Plutonium incorporated into the body is known to cause cancers, primarily at the sites of deposition that receive the largest radiation doses that include the lungs after inhalation of plutonium aerosols, and liver and skeleton after systemic distribution. Large epidemiology studies of the Mayak PA workers have established radiation dose-related cancers in these organs [4, 5]. While possible plutonium-associated pathologies were not established in the present study, it is possible that some of the identified chronic diseases that were associated with increased rates of plutonium excretion may have been caused or aggravated by the radiation exposures. Regardless of the etiology of the chronic diseases noted in this study, plutonium excretion rates and the distributions of plutonium between the liver and the skeleton were affected by the severity of the chronic diseases.

There are some limitations to this initial study that should be noted. The clinical histories were recorded over a period of 37 years (1971–2008) and there were considerable changes in clinical practices during this period. The interpretation of the clinical histories and records and subsequent assignment to the health status categories was based in clinical judgment of the available records and thus somewhat subjective. This initial study was also limited by the number of cases that could be identified that met the criteria for entry into the study that included suitable clinical histories, multiple bioassays over time, and suitable autopsy records. Additionally, the bioassays were conducted over a period of almost 4 decades. During this time, there were substantial improvements in the methodologies and instrumentation for both plutonium measurements in urine from bioassays and tissues from autopsy that included transitioning from alpha-radiometry in the earlier years to more sensitive alpha-spectroscopy in the later years [10, 21, 30].

This study shows that plutonium excretion rates measured at bioassay depend to a great measure on the health status of the individual at the time of the bioassay. If there were ongoing diseases that may affect the liver, this could cause significant deviations in the results of the bioassay measurements. Biokinetic models are used to estimate radiation doses to organs based on results from bioassays and autopsy data. If these models, such as ICRP-67 [31], assume a healthy individual, the bioassay measurements for workers with chronic diseases may overestimate the body and organ doses. Because increased severity of chronic diseases were also found to be associated with reductions in liver relative to skeletal contents of plutonium, the use of liver plutonium contents obtained after autopsy might underestimate radiation doses using standard biokinetic models. Thus, not considering the health status of the workers when the assays are conducted could lead to either over estimates or under estimates of disease risk from radiation exposures.

In conclusion, this initial study of Mayak workers shows that the severity of some chronic diseases known to affect the liver are associated with increased plutonium excretion rates and relative decreases in liver plutonium contents. Thus, the health status of the individual should be considered at the time of in-life bioassay and chronic and acute diseases should also be considered when autopsy data are used. Further studies are warranted to better determine the relationship of excretion enhancements with specific diseases and to the shorter-term consumption of alcohol on bioassay results and subsequent dose calculations and cancer risk estimates.

## Acknowledgments

The authors acknowledge the support and assistance of Barrett Fountos, former Program Manager, US Department of Energy, Office of Domestic and International Health Studies and Dr. Sergey Romanov, Director of the Southern Urals Biophysics Institute. This work was conducted as part of the Joint Coordinating Committee for Radiation Effects Research, Project 2.4, Mayak Worker Dosimetry. The authors express their appreciation for the many other professionals who worked over the years to provide the information used in this study.

## Author Contributions

**Conceptualization:** Klara G. Suslova, Alexandra B. Sokolova, Scott C. Miller.

**Data curation:** Alexandra B. Sokolova.

**Formal analysis:** Klara G. Suslova, Alexandra B. Sokolova, Scott C. Miller.

**Funding acquisition:** Alexander V. Efimov, Bruce A. Napier.

**Investigation:** Klara G. Suslova, Alexander V. Efimov, Alexandra B. Sokolova.

**Methodology:** Klara G. Suslova, Scott C. Miller.

**Project administration:** Alexander V. Efimov, Bruce A. Napier.

**Supervision:** Alexander V. Efimov, Bruce A. Napier.

**Writing – original draft:** Klara G. Suslova, Alexander V. Efimov, Alexandra B. Sokolova.

**Writing – review & editing:** Bruce A. Napier, Scott C. Miller.

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
