## [Decision Letter · Decision Letter 0]

10 Jul 2020

PONE-D-20-04522

The effects of chronic diseases on plutonium urinary excretion in former workers of the Mayak Production Association

PLOS ONE

Dear Dr. Napier,

Thank you for submitting your manuscript to PLOS ONE. After careful consideration, we feel that it has merit but does not fully meet PLOS ONE’s publication criteria as it currently stands. Therefore, we invite you to submit a revised version of the manuscript that addresses the points raised during the review process.

The reviewers raised several concerns, including missing experimental details (e.g., the plutonium analysis), that are minor but need to be addressed.  Reviewer 3 pointed out that “plutonium may have resulted in liver diseases, which then in turn reduces the storage of plutonium in the liver.” This possibility could at least be acknowledged in the Discussion section.

We look forward to receiving your revised manuscript.

Kind regards,

Hans-Joachim Lehmler, PhD

Academic Editor

PLOS ONE

Journal Requirements:

2. Please provide additional details regarding participant consent.

In the ethics statement in the Methods and online submission information, please ensure that you have specified (i) whether consent was informed and (ii) what type you obtained (for instance, written or verbal, and if verbal, how it was documented and witnessed).

If your study included minors, state whether you obtained consent from parents or guardians.

If the need for consent was waived, please ensure that you have discussed whether all data were fully anonymized before you accessed them and/or whether the IRB or ethics committee waived the requirement for informed consent.

3. To comply with PLOS ONE submission guidelines, in your Methods section, please provide additional information regarding your statistical analyses.

For more information on PLOS ONE's expectations for statistical reporting, please see https://journals.plos.org/plosone/s/submission-guidelines.#loc-statistical-reporting

Reviewers' comments:

Reviewer's Responses to Questions

**Comments to the Author**

1. Is the manuscript technically sound, and do the data support the conclusions?

Reviewer #1: Yes

Reviewer #2: Partly

Reviewer #3: No

Reviewer #4: Partly

2. Has the statistical analysis been performed appropriately and rigorously? 

Reviewer #1: Yes

Reviewer #2: Yes

Reviewer #3: No

Reviewer #4: Yes

3. Have the authors made all data underlying the findings in their manuscript fully available?

Reviewer #1: Yes

Reviewer #2: Yes

Reviewer #3: No

Reviewer #4: Yes

4. Is the manuscript presented in an intelligible fashion and written in standard English?

Reviewer #1: Yes

Reviewer #2: Yes

Reviewer #3: Yes

Reviewer #4: No

5. Review Comments to the Author

Reviewer #1: This study aimed to explore whether some chronic diseases, particularly those affecting liver, could enhance excretion rates of plutonium, having considered fourteen cases from former workers at the Mayak Production Association. The authors found that more serious chronic diseases, i.e., cardiovascular and liver diseases, were associated to increased plutonium excretion as well as with less retention of plutonium in the liver relative to that in the skeleton as determined after autopsy.

Overall, the manuscript is well written, methods and results are presented clearly, and discussion, including statement of study limits, is sufficiently detailed.

Minor comments.

Line 88. Based on Table 1, the workers finished working from 1956-1992 (not 1999), thus employment ranges from 7-33 years.

Table 1. Please amend the typo on 11th line, last column: 1985, not 1085.

Reviewer #2: The manuscript is very well interested and offer important way in the filed of environmental health. I have some recommendation for authors:

1. INTRODUCTION

The introduction part of the manuscript is very well written.

2. MATERIALS AND METHODS:

Please added the type of epidemiological study design; Period and area of observation.

Line 74: How did you define the criteria for cases selected (based on literature review, clinical recommendation, etc..). Please explain and put in the manuscript.

Line 79: Please define the "the different health categories". Added the explanation.

I recommended put the Tables 1, 2 and 3 in the results part of the manuscript.

Line 149: Please added the short description of methodology for Urine bioassays and autopsy tissue measurements for plutonium.

3. RESULTS

Line 232: Rewriten the title of table 5: "Correlation of systemic plutonium urinary excretion measured from in-life bioassays and relative plutonium distribution between liver and skeleton determined at autopsy for the three health groups of former Mayak PA workers.

4. DISSCUSION

General comments: rewritten the discussion part in this way:

1. The most important findings of your results related to the aim of the study.

2. Comparison your results with similar studies in this filed of research.

3. Limitation of results, methodology as well the strengths of your results and used methodology approach.

4. Why is your study important for the environmental health science.

5. Conclusion

Need to be shorter.

Reviewer #3: This study investigates plutonium activity in urine for the indirect estimation of doses of internal exposure; however, acknowledging that liver diseases, that could result from plutonium exposure could also affect the excretion of plutonium. This study of 14 cases shows the effect of chronic disease on plutonium excretion and the relative distribution of plutonium between the liver and the skeleton. Workers provided 4-9 urine bioassays for plutonium, had clinical records of their disease history, and had an autopsy conducted after death. Although there is repeated measurement data, the data analysis is cross-sectional. There is a need of proper biostatistical analyses.

Major Revisions

1. The authors analyze the data as cross-sectional observations, not using repeated assessments of plutonium (4-9 bioassays) nor the repeated disease status measurements.

2. The observation that they found relatively less plutonium in the liver relative to the skeleton determined by analyses conducted after autopsy may be due to their cross-sectional analysis.

3. The authors state in their abstract that plutonium excretion increased with more serious chronic diseases, including cardiovascular diseases and particularly diseases that involved the liver. However, they did not show any association with cardiovascular diseases in the main body of the text.

4. Although they have longitudinal data, the authors did not investigate whether plutonium may have resulted in liver diseases, which then in turn reduces the storage of plutonium in the liver. In the introduction they state that chronic diseases are associated with less retention of plutonium in the liver relative to that in the skeleton as determined after autopsy. However, the ‘causal link’ could likely be: “plutonium in the liver results in more liver diseases. The authors need to take the time order of the measurements of plutonium and the intermediate and final disease statuses into account (see Table 2). Hence in this paper, risk and consequences of these risks (diseases) are turned around. The final disease becomes the risk factor. See also Discussion section, page 13, line 243-247.

5. Page 10, Iine 173. It is not clear why the plutonium burden is extrapolated to an ‘ideal’ organ mass taking the ratio of actual organ mass and the organ mass from a reference man.

Minor Revisions

1. Page 6, 114-116: ‘also available’ repeated twice. Improve the style.

2. Page 8, Table 2: ‘Health Group’ should read ‘Health Status’

3. Page 10: GSD is not defined.

4. Pate 11, line 195-198 belongs to the Method section, not the Result section.

Same on page 12, line 214-215.

5. Page 14, line 255: “americium” needs to be capitalized.

6. Page 15: ‘ICRP 67’ is not defined.

Reviewer #4: The present manuscript reports a well-designed and structured work related with the excretion of plutonium from the human body and the prevalence of chronic diseases in workers from plutonium and radiochemical plants. The manuscript presents unique excretion rate profiles and health conditions of the selected workers which is relevant to evaluate their occupational exposure over an entire work career.

Please considerer some suggestions and recommendations to improve the manuscript.

Is there any information available in the literature reporting occupational exposure to plutonium thorough the inhalation? If so, authors could briefly describe that information and the major findings of those studies.

In the experimental section please provide a brief description of the methodologies used to determine plutonium in the urine and in organs. Please also add QA/QC data.

It would be interesting to some the major findings in a graphic comparison the three group of workers considered.

Lines 46-47: Authors should present the major possible sources of variability and highlight how the improved calculated methods reduce that variability.

Lines 58-59: If possible, authors could highlight the most described diseases, which would draw more attention to the relevance of this study.

Lines 177-180: this information could be integrated at the end of the previous sub-section.

Table 2,3, 5:

The column "Health Group" should be renamed as Group or similar to be in line with Table 4 which seems more correct since among the three groups considered only one refers to healthy workers. Authors should use a uniform way to mention the groups.

Table 4:

1) the geometric SD is predominantly higher than the respective geometric mean (except for the severely Ill), so this parameter may not be the most adequate to describe the profile of Plutonium excretion. Thus, it would be interesting if the range (minimum - maximum) of values observed in the participants was presented.

2) if the statistically significant differences were observed for all the three groups of workers, then the letter "a" should be in the column of the arithmetic mean (and not only in the line of health group).

Table 5

1) the letters "a" and "b" should be added to the Ill and Severely Ill lines of Ke syst.

2) The ratio liver/skeleton should be added to the table

Lines 281-289: Authors should highlight the potential of including workers' personal sampling to breathable plutonium in airborne particulate matter during regular working hours to a more complete health risk assessment.

Please see also some suggestions to improve the style and grammar used:

The expression "particularly those that affect the liver" are repetitively used in the text. I suggest the authors to adjust the manuscript t to become less repetitive.

line 27, 30-31: please rephrase to become clearer

lines 63-65: please rephrase this sentence

Line 89: The workers had worked (...); please rephrase.

6. PLOS authors have the option to publish the peer review history of their article (what does this mean?). If published, this will include your full peer review and any attached files.

Reviewer #1: **Yes: **Francesca Gorini

Reviewer #2: No

Reviewer #3: No

Reviewer #4: No

---

## [Author Response · Author response to Decision Letter 0]

25 Sep 2020

Response to Reviewers and additional information requested from the Editor. 

 We thank the 4 reviewers for their comments and support of our work. We have tried to incorporate the suggestions from the reviewers and have added additional clarifications, some additional references, and extended and clarified some of the discussion points. 

Editor Requests

1. Style requirements. We have complied with this request.

2. Participant consent and anonymization: We have now added this complete statement at the beginning of the Methods section. “The study was done using anonymized records and was approved by the Human Ethics Review Panel of the Southern Urals Biophysics Institute and in compliance with the laws of the Russian Federation.”

We have also indicated in the next sentence that the records had been “de-identified”; “ Fourteen de-identified cases were selected from the Mayak PA database…..”

3. Statistical Analyses. Additional information is now provided in the Methods sections.

4. Availability of the data. Access to the data is available only upon request in compliance with the “Data Access Agreement” from the Joint Committee on Radiation Effects Research (JCCRER). The JCCRER is a committee established by a bilateral agreement between the U.S. and Russian Governments, signed by the U.S. Secretary of State and the Foreign Minister for the Russian Federation. Under this agreement, requests for data can be directed to Dr. Sergey Romanov, Director of the Southern Ural Biophysics Institute, Ozyorsk, Russia at Romanov@subi.ru.

Responses to Reviewers. 

Reviewer #1: We appreciate the supportive comments from this reviewer. 

Minor comments: The minor errors were corrected. Thank you! 

Reviewer #2: We agree that this manuscript advances the field of environmental health, particularly for those either exposed or potentially exposed to some internally incorporated radionuclides and perhaps some other heavy metals. 

1. Introduction: The introduction part of the manuscript is very well written. 

 We have made some changes in the Introduction as suggested by the other reviewers. 

2. Materials and Methods: Please add the type of epidemiological study design; Period and area of observation.

 We have reorganized parts of the Materials and Methods and have added additional information that includes that the workers were observed as in-patients in the clinics where both health assessments and bioassays were conducted, thus the biophysical examinations correspond in time with the medical examinations (in Table 1). Some additional information was also provided in the statistical methods section that hopefully adds some clarity to the design of this retrospective study using work history, medical, bioassay and autopsy data. 

How did you define the criteria for cases selected (based on literature review, clinical recommendation, etc..). Please explain and put in the manuscript.

 We have added a new paragraph at the first of the Introduction that provides some general background on plutonium metabolism and toxicity. The criteria for selection of cases were determined based on the earlier studies [references #15, 16] and known other causes of chronic liver disease (such as congestive heart failure) as well as the availably of the necessary information and data to conduct the study. The necessary criteria are listed in the first part of the Materials and Methods and additional information is provided in this section, including additional references in this section and the Discussion on congestive heart failure. We do want to emphasize that while the entire Mayak Worker Cohort is quite large (about 26,000 workers), there are relatively few workers who fit the specific and necessary criteria defined in our manuscript. 

Line 79: Please define the "the different health categories".

 We have moved this sentence (former line 79) to the later section ‘Determination of health status’ (now line 114) where the health status categories are now defined and referenced in greater detail. 

I recommended put the Tables 1, 2 and 3 in the results part of the manuscript.

 These tables (1-3) mainly contain descriptive characteristics of cases whereas the excretion and autopsy data are the primary results (Table 4-5). Our preference is to leave the Tables in proximity to where they are first cited for the convenience of the reader. 

Please add the short description of methodology for Urine bioassays and autopsy tissue measurements for plutonium.

 The assays are now briefly described and references cited where the techniques are described in detail, see lines 158-163 for urine bioassays and the new section on Autopsy measurements, lines 165-174. 

3. Results: The title to Table 5 has been changed as suggested. 

4. Discussion: The Discussion has been reorganized as suggested with some additions to address the importance in this field and further discussion points raised by the Editor and other reviewers. 

5. Conclusion: We have shortened and focused the Conclusion as suggested.

Reviewer #3:

1. The authors analyze the data as cross-sectional observations, not using repeated assessments of plutonium (4-9 bioassays) nor the repeated disease status measurements.

 Additional information has been added to the Materials and Methods, including more information in the Statistics section. The reviewer is correct in that the data were analyzed cross-sectionally. While there were multiple bioassays done over time in each individual, the mean of the bioassay measurements were used for each worker in each health category (see revised Statistical Methods section). 

2. The observation that they found relatively less plutonium in the liver relative to the skeleton determined by analyses conducted after autopsy may be due to their cross-sectional analysis.

 The skeleton and liver contents were real measurements of Pu content done after autopsy and not calculated from biokinetic models from the in-life urine bioassay data. We should note that the estimated organ nuclide contents are commonly done in living individuals using various biokinetics models, but much of the foundation for these biokinetics models comes from actual measurements done at necropsy in animal studies or autopsies in human studies. One of the main purposes of the present study is to provide information on worker diseases that may influence the interpretation of these in-life bioassay measurements. 

3. The authors state in their abstract that plutonium excretion increased with more serious chronic diseases, including cardiovascular diseases and particularly diseases that involved the liver. However, they did not show any association with cardiovascular diseases in the main body of the text.

 We thank the reviewer for asking this question as we agree that this is a subject that is important and needs further emphasis, particularly since 4 of the 6 workers in the “severely ill” categories had also been diagnosed with congestive heart failure. The relationship of heart disease with liver disease and visa versa is recognized and we have provided an additional recent reference (ref #24) in the Discussion on the importance of the disease relationships between the liver and heart. The relationship of cardiovascular-liver disease and the retention of plutonium and other nuclides and metals is a topic of interest that we are going to pursue in future studies.

4. Although they have longitudinal data, the authors did not investigate whether plutonium may have resulted in liver diseases, which then in turn reduces the storage of plutonium in the liver. In the introduction they state that chronic diseases are associated with less retention of plutonium in the liver relative to that in the skeleton as determined after autopsy. However, the ‘causal link’ could likely be: “plutonium in the liver results in more liver diseases. The authors need to take the time order of the measurements of plutonium and the intermediate and final disease statuses into account (see Table 2). Hence in this paper, risk and consequences of these risks (diseases) are turned around. The final disease becomes the risk factor. 

 The purpose of this study was not to establish the etiologies of the livers diseases, rather to correlate disease severity with plutonium excretion. This is important to adequately estimate plutonium body content, calculate dose, and ultimately estimate cancers risks in living radiation workers or exposed members of the public. There were, however, indications of the etiologies of some of these diseases in the clinical records, such as congestive heart failure, metastatic disease and alcohol-related disease. However, the reviewer does have a valid point worthy of further discussion and clarification. Thus, we have added a paragraph in the Discussion (lines 288-298) that discusses the large epidemiological studies done in the Mayak Worker Cohort (>26,000 workers) and the excesses in cancers per absorbed radiation dose in the primary target organs that include the liver. Some of these references are also now included in the Introduction. We also raise the possibility, as noted by the reviewer, that radiation exposures may have contributed to some of the liver diseases, such as metastatic disease, that were noted in the clinical records. We should note that there are no biological/molecular markers that are definitive for Pu-induced cancers, thus the associations of radiation exposures and various diseases is a statistical exercise as presented in the epidemiological studies now cited in the revised Introduction and Discussion (new references #2-7). 

5. It is not clear why the plutonium burden is extrapolated to an ‘ideal’ organ mass taking the ratio of actual organ mass and the organ mass from a reference man.

 The historic autopsy protocols that were conducted at the Southern Urals Biophysics Institute for the workers of the Mayak Production Association did not include entire organ weights (which would almost be impossible for the entire skeleton, for example), rather only portions of organs for histopathological evaluations. Thus, the International Commission on Radiation Protection (ICRP) “Standard Man” was used for the liver and the skeleton mass was calculated as per the reference cited in the manuscript (reference #22). 

Minor Revisions: The identified minor revisions have all been addressed in the revised manuscript. We appreciate those suggestions. 

Reviewer #4: 

We appreciate the supportive comments of the importance of our work.

Is there any information available in the literature reporting occupational exposure to plutonium thorough the inhalation? If so, authors could briefly describe that information and the major findings of those studies.

 We appreciate this suggestion to help the general reader with this important topic related to radiation-induced cancers. We have added a new paragraph at the beginning of the Introduction as well as a new paragraph in the Discussion that presents some of this background with new citations where the interested reader will be directed to some of the current literature on this. 

In the experimental section please provide a brief description of the methodologies used to determine plutonium in the urine and in organs. Please also add QA/QC data.

 We now identify in the revised text in the Methods where the limitations, sensitivities, uncertainties and QA/QC of different assay techniques are described. Specifically, lines 160-163 and 168-174 in the Methods and additions to the paragraph on limitations, lines 299-309, in the Discussion. 

It would be interesting for some to have major findings in a graphic comparison the three group of workers considered.

 We considered different ways to present some of these data graphically, but felt that the presentation of these data in Table form were more informative and concise. 

Authors should present the major possible sources of variability and highlight how the improved calculated methods reduce that variability.

 This topic is very important and has been detailed extensively in prior publications, the most relevant of which are cited in this manuscript. We have revised the Methods (both Urine Bioassay and the new Autopsy measurements sections) as well as parts of the Discussion to help direct the reader to this literature. 

If possible, authors could highlight the most described diseases, which would draw more attention to the relevance of this study.

 We appreciate this suggestion and have addressed two aspects of this. First we have now included more background on cancers associated with plutonium exposures in human in the first paragraph of the Introduction. Additionally, we have provided additional information on the conditions (such congestive heart failure) associated with the observed liver diseases in the Discussion and a new recent reference (revised paragraph from lines 265-270).

Lines 177-180: this information could be integrated at the end of the previous sub-section.

 Even though this is a very short section, these data for the liver and skeleton are derived from autopsy data whereas the data in the prior section refer to the calculations of the urine excretion coefficients derived from in-life urine bioassays. We prefer to keep these separate to avoid misunderstandings of how these different data were derived. 

Tables: The column "Health Group" should be renamed. 

 The groups have now been consistently labeled and referred to as “Health Status Groups” in the Tables and the body of the manuscript. 

Table 4:

1) the geometric SD is predominantly higher than the respective geometric mean (except for the severely Ill), so this parameter may not be the most adequate to describe the profile of Plutonium excretion. Thus, it would be interesting if the range (minimum - maximum) of values observed in the participants was presented.

 We have now included the range in Table 4. 

2) if the statistically significant differences were observed for all the three groups of workers, then the letter "a" should be in the column of the arithmetic mean..

 Done. 

Table 5

1) the letters "a" and "b" should be added to the Ill and Severely Ill lines of Ke syst.

 Done

2) The ratio liver/skeleton should be added to table 5. 

 Done 

Lines 281-289: Authors should highlight the potential of including workers' personal sampling to breathable plutonium in airborne particulate matter during regular working hours to a more complete health risk assessment.

 Unfortunately, we do not have access to all of this information. However, we should mention that we have now included additional background information and references on some of the large epidemiological studies on the Mayak Worker Cohort where some aspects of the nature of the industrial aerosols resident in different plant locations and changes over time are considered. 

Please see also some suggestions to improve the style and grammar used:

 We have tried to improve the style and grammar as suggested. 

 We thank the reviewers for their constructive and helpful comments and we feel the manuscript has been greatly improved by these suggestions.

---

## [Decision Letter · Decision Letter 1]

28 Oct 2020

The effects of chronic diseases on plutonium urinary excretion in former workers of the Mayak Production Association

PONE-D-20-04522R1

Dear Dr. Napier,

We’re pleased to inform you that your manuscript has been judged scientifically suitable for publication and will be formally accepted for publication once it meets all outstanding technical requirements.

Kind regards,

Hans-Joachim Lehmler, PhD

Academic Editor

PLOS ONE

Reviewers' comments:

Reviewer's Responses to Questions

**Comments to the Author**

1. If the authors have adequately addressed your comments raised in a previous round of review and you feel that this manuscript is now acceptable for publication, you may indicate that here to bypass the “Comments to the Author” section, enter your conflict of interest statement in the “Confidential to Editor” section, and submit your "Accept" recommendation.

Reviewer #2: All comments have been addressed

Reviewer #4: All comments have been addressed

2. Is the manuscript technically sound, and do the data support the conclusions?

Reviewer #2: Yes

Reviewer #4: Yes

3. Has the statistical analysis been performed appropriately and rigorously? 

Reviewer #2: Yes

Reviewer #4: Yes

4. Have the authors made all data underlying the findings in their manuscript fully available?

Reviewer #2: Yes

Reviewer #4: Yes

5. Is the manuscript presented in an intelligible fashion and written in standard English?

Reviewer #2: Yes

Reviewer #4: Yes

6. Review Comments to the Author

Reviewer #2: I AGREE WITH AUTHORS CORRECTIONS

Reviewer #4: The manuscript was greatly improved after the revision. All suggestions and clarifications were addressed.

7. PLOS authors have the option to publish the peer review history of their article (what does this mean?). If published, this will include your full peer review and any attached files.

Reviewer #2: **Yes: **Andreja Kukec

Reviewer #4: No

---

## [Editor Report · Acceptance letter]

5 Nov 2020

PONE-D-20-04522R1 

The effects of chronic diseases on plutonium urinary excretion in former workers of the Mayak Production Association 

Dear Dr. Napier:

I'm pleased to inform you that your manuscript has been deemed suitable for publication in PLOS ONE. Congratulations! Your manuscript is now with our production department. 

Kind regards, 

on behalf of

Dr. Hans-Joachim Lehmler 

Academic Editor

PLOS ONE